# Impact of the COVID-19 Pandemic on Lifestyle Behavior and Clinical Care Pathway Management in Type 2 Diabetes: A Retrospective Cross-Sectional Study

**DOI:** 10.3390/medicina60101624

**Published:** 2024-10-04

**Authors:** Giovanni Cangelosi, Stefano Mancin, Paola Pantanetti, Marco Sguanci, Sara Morales Palomares, Alessia De Luca, Federico Biondini, Francesco Tartaglia, Gaetano Ferrara, Fabio Petrelli

**Affiliations:** 1Unit of Diabetology, Asur Marche—Area Vasta 4 Fermo, 63900 Fermo, FM, Italy; giovanni.cangelosi@virgilio.it; 2IRCCS Humanitas Research Hospital, 20089 Rozzano, ML, Italy; stefano.mancin@humanitas.it; 3A.O. Polyclinic San Martino Hospital, Largo R. Benzi 10, 16132 Genova, GE, Italy; sguancim@gmail.com; 4Department of Pharmacy, Health and Nutritional Sciences, University of Calabria, 87036 Rende, CS, Italy; sara.morales@unical.it; 5School of Biosciences and Veterinary Medicine, University of Camerino, 62032 Camerino, MC, Italy; alessia.deluca092@gmail.com; 6Unit of Psychiatry, Ast Fermo, 63900 Fermo, FM, Italy; federico.biondini@sanita.marche.it; 7Department of Biomedical Sciences, Humanitas University, 20072 Pieve Emanuele, ML, Italy; francesco.tartaglia@st.hunimed.eu; 8Nephrology and Dialysis Unit, Ramazzini Hospital, 41012 Carpi, MO, Italy; amaranto1984@libero.it; 9School of Pharmacy, Polo Medicina Sperimentale e Sanità Pubblica “Stefania Scuri”, 62032 Camerino, MC, Italy; fabio.petrelli@unicam.it

**Keywords:** type 2 diabetes (T2D), COVID-19, PACIC Questionnaire, Medi-Lite Questionnaire, social restriction

## Abstract

*Background and Objectives:* In Italy, as in the rest of the world, government restrictions aimed at containing the spread of COVID-19 primarily imposed limitations on social relationships and personal behavior. This situation significantly affected the management of chronic illnesses, including type 2 diabetes (T2D). The objective was to evaluate the perceptions of patients with T2D regarding the quality of care received during the COVID-19 pandemic and the impact on dietary and physical activity behaviors. *Materials and Methods*: We conducted a retrospective cross-sectional survey. Data were collected from June to July 2023 using the convenience sampling of patients with T2D, and the Patient Assessment of Chronic Illness Care (PACIC) and Medi-Lite questionnaires were administered. *Results:* During the research period, out of the 130 subjects who met all enrollment criteria, 103 patients were included in this study (79.23%). The results of the administered questionnaires were heterogeneous. The average scores from the PACIC Questionnaire for each question displayed significant variability, indicating a range of experiences in the quality of care. In the Medi-Lite survey, fruit, cereals, and olive oil showed the highest adherence levels, with mean scores ranging from 2.58 (SD ± 1.18) for fruit to 1.89 (SD ± 0.34) for olive oil and 1.97 (SD ± 0.17) for cereals. Patients who reported increased food intake during the lockdown attributed it to having more time to prepare meals. Physical activity levels remained unchanged for 48 patients, decreased for 45 patients, and only 9 patients managed to exercise more during the COVID-19 restrictions. *Conclusions:* Healthcare systems must prioritize comprehensive care plans for T2D that address not only physical health, but also emotional and social well-being. Post-pandemic, promoting healthier lifestyles and empowering patients to manage their condition is crucial. A multidisciplinary and multidimensional approach could support the care of vulnerable individuals, such as patients with T2D, especially during crises like pandemics or other dramatic events.

## 1. Introduction

The COVID-19 pandemic, declared by the World Health Organization (WHO) on 11 March 2020 and concluded on 5 May 2023, resulted in more than seven hundred million infections and possibly seven million deaths (as of 13 April 2024). Over about four years since its global spread, it reached dimensions that posed significant challenges for public and social health management in recent history [1,2]. Several studies have addressed the impact of COVID-19 on different populations, focusing on predicting rapid disease progression and long-term consequences. Persistent lung damage was observed in some patients even after two months, with long-term effects examined up to two years post-infection. These studies revealed that patients with pulmonary fibrosis experienced ongoing proinflammatory and prothrombotic conditions. Elevated D-dimer levels were linked to reduced lung function, highlighting potential therapeutic and prognostic implications [3,4,5]. In Italy, as in the rest of the world, government restrictions to contain the spread of COVID-19 imposed limitations primarily on social relationships and personal behavior, with the initial period of restrictions occurring between 9 March and 18 May 2020 [6,7]. Diabetes and cardiovascular disease, with obesity as a major comorbidity, were identified early on as significant risk factors for developing severe conditions from COVID-19 [8,9,10,11,12,13,14,15]. Diabetes, particularly type 2 diabetes (T2D), has always been a condition that significantly influences the lifestyle habits of patients, even before the SARS-CoV-2 pandemic [16,17,18,19]. Existing research indicates that the COVID-19 pandemic led to a significant decrease in emergency surgical procedures and a reduction in overall emergency department visits, primarily due to the widespread fear and anxiety many patients experienced about hospitals during the height of the outbreak [20]. Diabetic patients, in particular, faced unique challenges during the pandemic, including increased delays in emergency care and heightened health risks, underscoring the need for targeted interventions [21]. As anticipated, the COVID-19 pandemic affected many chronic diseases to varying degrees; however, there is insufficient evidence to identify the most vulnerable patient groups [22,23]. Diabetes affects 425 million people worldwide (ages 20–79), with 87% to 91% suffering from T2D, and 79% living in low- to middle-income countries. At this uncontrolled rate of growth, the number of affected individuals is projected to increase to 629 million by 2045, with an overall cost to health institutions of around USD 727 billion [16,24,25,26]. T2D is one of the most common chronic diseases globally and one that involves a high risk of permanent disability [27]. In Italy, in 2023 (Istituto Superiore di Sanità, ISS), almost 4 million citizens are affected by diabetes (approximately 5.3% of the total population; among those, 16.5% are over 65 years of age); it has also been observed that 28.9% of diabetics are obese and sedentary [28]. These significant epidemiological factors make diabetes and its management (particularly T2D) one of the main global challenges when considering the period starting from the early stages of the COVID-19 pandemic and throughout the four years of the most severe health emergency recorded in the last century [29,30,31,32,33]. This period has highlighted significant challenges in healthcare, organizational, and clinical aspects, as several studies have also revealed [34,35,36,37,38,39]. Evidence from epidemiological studies suggests that weight gain, lack of physical exercise, and lifestyles characterized by stress, smoking, and alcohol consumption play crucial roles in the onset of T2D. Overweight or obese conditions [Body Mass Index (BMI) of 25 to 30 or higher] are strongly associated with insulin resistance [16,17,24,26,27,28,29]. During COVID-19 restrictions, particularly during lockdowns, the management of chronic diseases and related conditions, as well as the adherence to healthy lifestyles, faced significant difficulties in social, organizational, and psychological aspects [40,41,42,43,44,45,46,47,48,49,50].

Some studies [51,52,53] have shown a significant increase in complications for patients with poorly managed T2D during pandemics, underscoring the importance of promoting healthy lifestyles and strategies to reduce psychological stress in these patients, particularly among the elderly, as evidenced by a qualitative research study [54].

The remote management of chronic conditions has been introduced as a tool in healthcare pathways to facilitate the management of complex clinical conditions. For example, a study [55] evaluated the effectiveness of a social media platform, LINE, in improving knowledge, attitudes, and self-management practices among T2D patients. This study highlighted the potential of such solutions to address the challenges posed by the pandemic, emphasizing the need for innovative approaches to ensure continuity of care for chronic patients, including those with T2D.

The lockdown due to the COVID-19 pandemic, despite its challenges in managing therapeutic interventions, offered, in some cases, an unexpected opportunity to improve lifestyle habits. A study conducted in France [56] revealed that a significant number of people with diabetes and who were overweight used this period to adopt a healthier lifestyle, lose weight, and improve their diet. This study evaluated T2D patient perceptions of care during COVID-19 and assessed their adherence to the Mediterranean diet and behavior changes, highlighting how environmental behaviors can be influenced by tragic events like the COVID-19 pandemic. It also emphasized the need for an integrated healthcare system that can respond effectively, even during prolonged catastrophic situations. A limitation of our study is the potential for recall bias, a common issue in retrospective studies as it relies on participant memory of past events, which can lead to inaccuracies and affect the reliability of the data.

## 2. Materials and Methods

### 2.1. Study Objectives

The primary objective of this study was to evaluate the perception of patients with T2D regarding the quality of care they received during the COVID-19 pandemic. The secondary objective was to assess adherence to the Mediterranean diet and the impact of COVID-19 on dietary and physical activity behaviors.

### 2.2. Study Design

A retrospective cross-sectional survey was conducted to assess the impact of the COVID-19 pandemic on the management of T2D among patients attending the Diabetology and Endocrinology Department of a tertiary hospital in Fermo, Marche Region, Italy. This study was reported in accordance with the Strengthening the Reporting of Observational Studies in Epidemiology (STROBE) Statement (Appendix A) [57].

### 2.3. Ethical Considerations

This study adhered to the principles outlined in the Declaration of Helsinki. All participants were informed about the study’s objectives, and consent was obtained in compliance with all privacy regulations (Art. 13 of EU Regulation 679/2016) before the survey was administered. The data were processed anonymously. Ethical approval was granted by the Institutional Review Board of Ast Fermo with authorization code INF02/2022, dated 1 December 2022.

### 2.4. Study Population

Participation in the study was voluntary. The sample consisted of patients from the Diabetology and Endocrinology Department of the Augusto Murri Hospital in the province of Fermo, Italy. Participants were enrolled between June and July 2023. The inclusion criteria were as follows: having been diagnosed with T2D at least two years before the COVID-19 pandemic; being in regular follow-up at the sampling center; being of either gender; aged 18 or over; having had signed informed consent and privacy protection forms; and being able to understand the questionnaires independently or with the support of a caregiver or health worker. The exclusion criteria included a diagnosis of major psychiatric disorders and participation in other clinical studies.

### 2.5. Data Collection

The data for this study were retrospectively extracted from electronic health records (EHRs) following a formal request sent to the hospital’s health administration. These data included information on changes in dietary habits and physical activity levels during the COVID-19 period. The data were collected at a single time point by a data manager who was not involved in this study and were processed in aggregate form. Socio-demographic data, such as gender, age, education level, type of employment, and the number of individuals in the household, were also extracted from the medical records. These data allowed us to stratify the population, make inferences based on the collected data, and study the phenomenon in greater depth.

### 2.6. Instrument

This study utilized validated Italian-language questionnaires to collect additional information, with assistance provided by caregivers or family members, and an operator was always available for clarification.

The primary instrument used was the Italian version of the Patient Assessment of Chronic Illness Care (PACIC) Questionnaire (Appendix A, both Italian and English versions) [58]. The PACIC Questionnaire, aligned with the principles of the Chronic Care Model (CCM) adopted by the study center [59,60], evaluates the comprehensiveness of social assistance, organizational principles, and self-care for chronic conditions. It includes 26 questions divided into five subscales: (a) patient activation, (b) delivery system design/decision support, (c) goal setting/tailoring, (d) problem-solving/contextual counseling, and (e) follow-up/coordination. Responses are measured on a Likert scale from 1 (“almost never”) to 5 (“almost always”), reflecting the care provided to patients with chronic diseases. The Medi-Lite Questionnaire (Appendix A, both Italian and English versions) [61,62], consisting of 9 questions, evaluates dietary habits by assessing the frequency of consumption of specific food groups: fruits, vegetables, legumes, cereals, fish, meat, dairy, alcohol, and olive oil. Response options range from “never or rarely” to “once or more times daily,” with scores assigned as 2 for the highest consumption category, 1 for the middle, and 0 for the lowest. The total score ranges from 0 to 18, indicating the level of adherence to the Mediterranean diet, with higher scores representing greater adherence. In addition to questionnaire responses, retrospective data were retrieved from electronic health records (EHRs) to complement the survey findings. The analysis specifically focused on the frequency of intake for various food groups and variations in physical activity levels compared to the period before 2020–2022.

### 2.7. Statistical Analysis

The statistical analyses for the PACIC and Medi-Lite questionnaires were conducted using Python, focusing on both descriptive and inferential statistics. The analysis of PACIC scores, which was undertaken to assess the patients’ perceptions of the care they received for chronic illness management, was subdivided according to the 5A subscales: Assess, Advise, Agree, Assist, and Arrange. Descriptive statistics, such as the mean and standard deviation (SD), were calculated for the PACIC items. These statistics provided a comprehensive overview of the central tendency and variability of responses within the entire sample and across stratified groups. Similarly, the Medi-Lite scores, which reflect adherence to the Mediterranean diet, were analyzed by stratifying the dataset according to demographic variables. For each stratification variable (e.g., gender and education level), the counts of Medi-Lite adherence levels (categorized as 0 [Low], 1 [Moderate], and 2 [High]) were computed for each dietary item (e.g., vegetables, legumes, and cereals). For both questionnaires, inferential statistical tests were performed to explore the relationships between the demographic factors and overall scores.

To explore the statistical correlations and assess the significance of differences between groups, the *t*-test was used, and the non-parametric Mann-Whitney U test was applied when the assumption of normality was not met for the distribution of the two samples.

## 3. Results

### 3.1. Characteristics of the Sample

Initially, 130 patients attending routine visits were considered. Of these, 20 did not meet the inclusion criteria. Among the 110 eligible patients, 5 did not give consent for the processing of personal data for scientific purposes. Out of the 105 participants who started in this study, 2 did not fully complete the questionnaires. Therefore, the final sample included in our study consisted of 103 patients (all Caucasian), which represents 79.23% of the initial sample (Figure 1).

The study population comprised 73 men and 30 women (*n* = 103), with mean ages of 68 and 67 years, respectively. The youngest patient was 33 years old, and the oldest was 89 years old (both males). The average number of family members per participant was 2.69. There were no significant differences between the two sexes in terms of educational qualifications (elementary, middle school, high school, or degree). The sample was equally divided between workers and pensioners, with 8.73% being unemployed (Table 1).

### 3.2. PACIC Questionnaire

Table 2 summarizes the general data collected from the cohort of subjects who gave consent to participate in the PACIC survey.

The mean scores obtained from the PACIC Questionnaire for each question showed considerable variability, reflecting diverse experiences in the quality of care. The PACIC questionnaire highlights a dichotomy in healthcare provision: while patients reported strong organizational support and adequate assistance with self-management, significant gaps existed in the patient involvement in decision making and the provision of external resources like support groups or local programs.

The items with the lowest average scores were “I have been given a copy of my treatment plan” (Item 9, mean score: 1.45 ± 1.12); “It was explained to me that my visits to other specialists (ophthalmologist, surgeon, etc.) have improved the treatment of my disease” (Item 19, mean score: 1.59 ± 1.26); and “I was asked how my visits to other specialists were proceeding” (Item 20, mean score: 1.73 ± 1.34). Conversely, the items with the highest average scores were “I have been asked to talk about the goals I have set for the treatment of my disease” (Item 7, mean score: 4.19 ± 1.42); “I have been given a written list of things to do to improve my health” (Item 4, mean score: 4.20 ± 1.19); and “I have been helped to develop plans to understand how to get support from friends, family, or the community in which I live” (Item 23, mean score: 4.37 ± 1.31).

A review of response frequencies shows that “almost never” was frequently selected for questions related to patient involvement, such as “I was given several treatment options to think about” (Item 2); “I was asked to talk about any problems I had with medications or their effects” (Item 3); and “I was asked what aspects of my illness I wanted to discuss during the visit” (Item 21).

Conversely, patients expressed positive regard for the hospital and the doctors and nurses who cared for them, as indicated by the majority of “almost always” responses to the items: “I could see that the care I was given was well organized” (Item 5) and “I was confident that, in recommending appropriate therapies, my doctor or nurse took into account my values and traditions” (Item 12).

The summary statistics for the 5As reveal insightful trends regarding the quality of care provided to patients based on their responses to the PACIC Questionnaire (Appendix A).

In the overall summary, the mean score across all respondents was 2.74, with a standard deviation of 0.60, indicating moderate variability in the patients’ assessments of care. The “Agree” component, which involves setting goals in partnership with the patient, had the highest mean score (3.39 ± 0.92), suggesting that patients felt more engaged in setting goals. In contrast, the “Assess” component, which involves evaluating the patient’s health status, had the lowest mean score (2.35 ± 0.78), indicating a need for more thorough assessments.

Data analysis also revealed some differences by gender. The male sample had a mean summary score of 2.65 with a standard deviation of 0.55, which was slightly below the overall mean. “Agree” was the highest scoring component (3.32 ± 0.90) while “Assess” was the lowest (2.26 ± 0.71). Females reported a higher overall mean score (2.96 ± 0.69) compared to men. They rated “Agree” the highest (3.57 ± 0.97) and “Assess” similarly low (2.58 ± 0.89), but generally scored higher across all components, suggesting a more positive perception of the care they received.

Aspects of educational qualification also showed variability by grade and employment status. The High School Graduates subgroup had an overall 5A summary score of 2.63 with a standard deviation of 0.56; “Agree” (3.25 ± 0.91) was the highest-rated component; and “Assess” (2.25 ± 0.77) was the lowest. The Degree Holders subgroup reported a higher overall score (3.28 ± 0.77), with “Agree” being the most positively rated (4.08 ± 0.90). This group’s scores were generally higher across all components, indicating a better assessment of care. The Secondary School Graduates subgroup had an overall score close to the overall mean (2.70 ± 0.65), with “Agree” being the highest-scoring component (3.30 ± 1.02). Similarly, in the Primary School Graduates subgroup, the overall score was slightly higher (2.83 ± 0.47), with “Agree” also being the highest-rated component (3.60).

The retired patients had an overall score of 2.66 with a standard deviation of 0.60. The “Agree” component was rated the highest (3.35 ± 0.81), while the “Assess” component was rated the lowest (2.28 ± 0.73), reflecting similar trends to other groups but with slightly lower overall satisfaction. Similarly, the employed subgroup scored slightly higher than the overall mean (2.79 ± 0.58), with “Agree” (3.39 ± 1.01) and “Assist” (3.08 ± 0.81) being particularly well rated, indicating positive experiences with setting goals and assistance. The unemployed subgroup showed an even higher mean score (2.91 ± 0.69), with “Agree” (3.55 ± 1.08) and “Assist” (3.07 ± 0.99) also rated highly, demonstrating relatively better satisfaction in these areas.

The 5A scoring data indicates that, while patients generally feel positively about the goal-setting aspect of their care, there are areas needing improvement, particularly in the “Assess” and “Advise” components. Females, degree holders, and the unemployed tended to rate their care experiences more positively, suggesting a need for more personalized care approaches.

### 3.3. Medi-Lite Questionnaire

Table 3 summarizes the general data collected from the cohort of subjects who gave consent to participate in the Medi-Lite survey.

Fruit, cereals, and olive oil were the components with the highest adherence, with mean scores ranging from 2.58 (SD ± 1.18) for fruit to 1.89 (SD ± 0.34) for olive oil (cereals: 1.97 ± 0.17). Other products received lower scores, indicating a low familiarity with the Mediterranean diet, particularly in the consumption of vegetables, legumes, and fish. On the positive side, the low intake of animal products is encouraging. However, the average score of 1.34 (SD ± 0.59) for alcohol consumption (which, in the Italian tradition, is often associated with wine) is somewhat controversial.

### 3.4. Impact of COVID-19 on the Dietary and Physical Activity Behaviors in Patients with T2D

Data extracted from clinical records and reports have highlighted a substantial shift in dietary and physical activity behaviors as a result of the lockdown. Increased time at home and boredom has led to higher food consumption, while movement restrictions and concerns about infection have limited physical activity. These findings indicate that pandemic-related restrictions have induced complex behavioral changes influenced by psychological and environmental factors (Figure 2).

#### 3.4.1. Changes in Dietary Habits

Data from clinical records and reports have revealed significant changes in dietary and physical activity behaviors due to the lockdown. Increased time spent at home and feelings of boredom have led to greater food consumption, while movement restrictions and concerns about infection have reduced physical activity. These findings suggest that pandemic-related restrictions have resulted in complex behavioral changes driven by psychological and environmental factors (Figure 2).

Patients who increased their food intake during the lockdown attributed this to having more time to prepare meals, such as leavened foods (e.g., bread and pizza) or foods with long preparation times (e.g., desserts), as well as meat, cold cuts, fruit, and sugary drinks. Four patients reported eating more due to concerns over the pandemic or boredom during the first lockdown, with increased consumption of sweets, pasta, leavened products, meat, cold cuts, and cheeses. One patient increased meat consumption due to greater economic availability, while two patients introduced more vegetables into their diet by starting self-production, which was facilitated by the extra time available during social restrictions. It was noteworthy that 13 of the patients could not identify a specific reason for their increased food intake. In these cases, the most consumed foods were sweets, fruit, cold cuts, and dairy products, which were mostly consumed outside main meals.

#### 3.4.2. Changes in Physical Activity Levels

Physical activity levels remained constant for 48 patients, decreased for 45 patients, and only 9 patients were able to exercise more during the COVID-19 restrictions. Those who decreased their activity levels did so primarily due to government restrictions that prevented them from going out or an increase in sedentary work. Moreover, 8 patients reduced their activity out of concern for contracting COVID-19, while 5 patients were unable to exercise because they had contracted COVID-19 themselves. All patients who increased their physical activity attributed their motivation to having more free time due to a lack of work. Constant levels of physical activity were maintained by 42 patients who continued their hobbies as before, as well as by 6 patients who continued to work in person during the most severe period of the pandemic. These findings underscore the significant impact of the COVID-19 pandemic on the dietary and physical activity behaviors among patients with T2D, highlighting the necessity for tailored interventions to support this vulnerable population during such crises (Figure 3).

### 3.5. Data Correlations

The statistical analysis conducted in this study identified several significant correlations between the variables considered and personalized care, as assessed through the PACIC and Medi-Lite questionnaires. These findings are particularly noteworthy for their implications in clinical practice, highlighting the potential for tailoring care strategies based on demographic factors such as gender, education level, and household size. They provide valuable insights that could enhance the effectiveness of personalized healthcare interventions (Appendix A).

#### 3.5.1. PACIC Questionnaire

The PACIC Questionnaire revealed statistically significant differences between the analyzed groups.

##### Gender

Women reported a significantly higher average score in the overall “5 A’s” summary compared to men (*t*-test, *p* = 0.015), suggesting that women perceive the overall quality of care more positively. Additionally, women significantly outscored men in the “Arrange” component, which assesses the perceived support in organizing follow-up care or referrals (Mann-Whitney U test, *p* = 0.022). This result indicates that women feel more supported in managing the continuity of care.

##### Education Level

Individuals with higher education levels reported a significantly higher “5 A’s” summary score compared to those with lower education levels (*t*-test, *p* = 0.030), suggesting that they perceive a better quality of care. In the “Assess” component, which evaluates the perception of health status evaluation, a significant difference was observed in favor of individuals with higher education (*t*-test, *p* = 0.047). This implies that they feel their health status is assessed more thoroughly.

In summary, women and individuals with higher education levels perceive better overall quality of care and feel more supported in follow-up processes and health status evaluations compared to their respective counterparts.

#### 3.5.2. Medi Lite Questionnaire

The Medi-Lite Questionnaire identified significant differences in various dietary consumption patterns, particularly in relation to gender, education level, and household size.

##### Gender

Alcohol consumption differed significantly between men and women (Mann-Whitney U test, *p* = 0.015), indicating distinct patterns of alcohol intake across genders. Dairy product consumption approached statistical significance (*p* = 0.05), suggesting a potential trend toward gender differences, although it did not reach the conventional threshold for statistical significance. No significant differences were observed in the consumption of vegetables, legumes, cereals, fish, meat, olive oil, or the overall dietary score between the two sexes.

##### Education Level

Significant differences were found in the consumption of cereals (Mann-Whitney U test, *p* = 0.021), fish (*p* = 0.040), and alcohol (*p* = 0.039), indicating that individuals with higher education levels exhibit distinct dietary patterns compared to those with lower education levels.

##### Household Size

Olive oil consumption varied significantly based on household size (Mann-Whitney U test, *p* = 0.005), with larger households consuming olive oil differently than smaller ones. The overall dietary pattern score showed a highly significant difference between households with fewer members and those with more members (*p* < 0.001), highlighting the substantial role household size plays in shaping overall dietary habits.

## 4. Discussion

This study aimed to explore the influence of the COVID-19 pandemic on the management of T2D, focusing on dietary and lifestyle factors. Our findings highlight the significant challenges and difficulties in managing T2D in an extremely vulnerable cohort during the pandemic. Even with first-choice pharmacological interventions, preventing more serious health consequences proved challenging for patients with T2D during this delicate period [63]. The multifactorial emotional vulnerability of these individuals likely affected both clinical and social outcomes during the global crisis [64]. The post-COVID-19 period has brought to light issues related to the direct management of care and the psychological state of patients with T2D, as well as the healthcare workers involved in their specialized care [65]. During the pandemic, managing T2D became more challenging due to disrupted care and lifestyle changes. Limited access to healthcare and increased stress contributed to worsened glucose control. An integrated and flexible care system, supported by continuous digital tools, is crucial for effective management [66,67].

Healthier lifestyles and proper diets, which have been long recommended for better management of diabetic and frail populations, were significantly impacted by the pandemic, affecting the ability to maintain satisfactory overall health conditions [68]. Our study primarily focused on these aspects but also sought to provide insights into better organizational management in the post-COVID-19 era [69]. This study found strengths in chronic care organization but gaps in patient involvement and external support. Dietary adherence was partial, with the lockdown impacting eating and activity [70,71]. T2D management varied by gender and education, with women and more educated individuals perceiving better care, possibly due to greater awareness and adherence [72].

These hypotheses are also supported by a recent qualitative study by Shi et al. [73], in which it was argued that low levels of education and advanced age limit individuals’ ability to acquire medical information. Similarly, a cross-sectional study demonstrated a significant relationship between gender and education status and the glycemic control of participants [74]. Additionally, women often seek support more actively and participate more in decisions about their health, even though, culturally, there are still significant global disparities in gender differences in disease and its related socio-health consequences. Improving health education and promoting gender equity could further enhance the results achieved and better tailor the provided care [75].

Regarding the Mediterranean diet, the variables that have shown a higher adherence to this diet are consistently higher education levels and the presence of more individuals within the same household. This finding aligns with two studies that describe how living alone is associated with financial, social, lifestyle, and environmental factors, which may influence dietary behaviors [76,77]. In this context, another review showed that individuals who live alone may have lower and insufficient intakes of some key foods, including fruits, vegetables, and fish, compared to those who do not live alone. This may lead to chronic diseases, such as T2D [78]. This study also revealed that men have a greater tendency to consume alcohol, as supported by a recent cohort study on the prevalence of alcohol consumption and its relationship with T2D [79]. Excessive alcohol consumption can negatively affect blood glucose levels, causing sudden fluctuations in blood sugar and contributing to an increase in BMI, which elevates the risk of developing complications related to T2D [80,81].

For more effective and personalized care, managing individuals with T2D must prioritize adopting the healthiest possible dietary approach, particularly in terms of “Mediterraneanness”, which means a diet rich in fruits, vegetables, whole grains, legumes, olive oil, and fish. This dietary pattern can improve outcomes in patients with diabetes by stabilizing blood glucose, reducing insulin resistance, and lowering BMI, thereby decreasing the risk of cardiovascular and metabolic complications [82,83]. These findings are consistent with previous studies, such as the one that examined the impact of COVID-19 on diabetic patients and revealed a significant decline in medication adherence and healthy lifestyle behaviors following lockdowns [84]. The necessity of the pandemic has paved the way for telemedicine assistance in managing chronic and mental illnesses from the early stages of the emergency, which could support treatment during stressful periods [85]. The introduction of telemedicine for managing these patients could have a positive impact, as demonstrated by studies showing significant improvements in psychological well-being and management of disease-related stress [86]. Although several studies have highlighted the benefits of telemedicine in diabetes education, some limitations have been identified, such as differences in digital literacy and access to technology among patients [87]. Future challenges will involve developing alternative and integrated programs for diabetes care management that place the patient at the center, involving a multidisciplinary team [88,89]. Telemedicine should be considered not only as a crisis-time solution, but also as a core strategy integrated into standard care [90]. This study aimed to preliminarily assess the community’s needs in the reference area to personalize the care of patients with T2D as effectively as possible. In line with our findings and previous research, it is imperative that healthcare organizations embrace telemedicine to enhance remote care for these patients. This can be achieved through the implementation of training programs and introductory practices that facilitate proper telemedicine utilization.

### Limitations

Our study has several limitations. First, it is a retrospective analysis of a limited sample of individuals with potentially compromised health behaviors prior to the COVID-19 pandemic. As a retrospective study, recall bias and the difficulty of accurately recalling past events must be considered. Additionally, this study was monocentric, conducted at a single center, and the sample was not stratified based on gender (the number of women participating in this study was much lower than men), clinical parameters, and lifestyle habits, which may have influenced the interpretation and generalization of the results. The absence of a control group also limited the ability to compare how the pandemic has affected care. Future studies with a more robust methodology, including comparisons between stratified patient groups, are recommended to obtain more precise information on chronic care and the impact of social restrictions during the pandemic era. Nevertheless, this study provides valuable information to researchers and the scientific community.

## 5. Conclusions

The results of this study demonstrate significant correlations between certain demographic variables and both the perception of care quality and dietary habits. Specifically, women and individuals with higher education levels tended to perceive better healthcare quality. Similarly, gender, education level, and household size significantly influenced dietary consumption patterns. These findings provide valuable insights for the personalization of care, suggesting that demographic factors can have a considerable impact on both the perception of healthcare and dietary choices. This study also highlighted the critical need for a comprehensive, multidisciplinary approach to managing chronic diseases. This approach should extend beyond direct professional treatment to include initiatives that promote healthy lifestyles and active patient participation. Furthermore, the findings emphasize the importance of developing tailored interventions that address the negative impacts of crises, such as lockdowns, on the dietary habits and physical activity of individuals with chronic conditions. Additionally, this study underscores the need for multidisciplinary interventions to support patients during challenging times, promoting treatment adherence through a holistic approach. By integrating healthcare professionals from various fields, we can create a robust framework that not only effectively manages the disease, but also empowers patients to lead healthier lives.

## Figures and Tables

**Figure 1 medicina-60-01624-f001:**
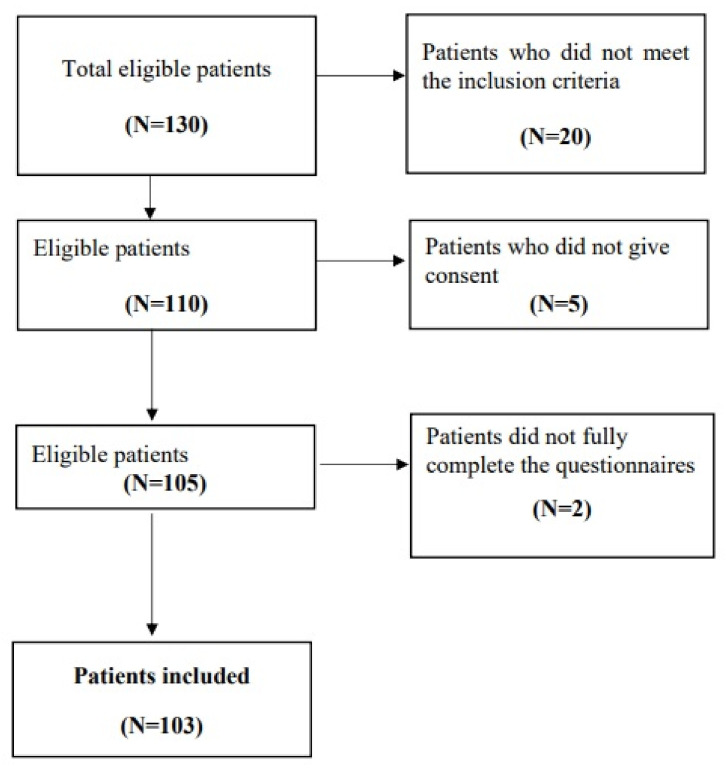
Sample recruitment flow chart.

**Figure 2 medicina-60-01624-f002:**
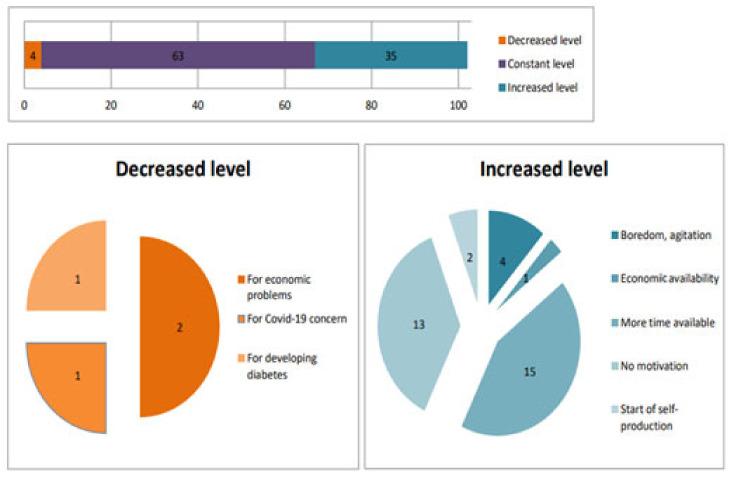
Qualitative questionnaire responses about the frequency of food intake levels during the COVID-19 period.

**Figure 3 medicina-60-01624-f003:**
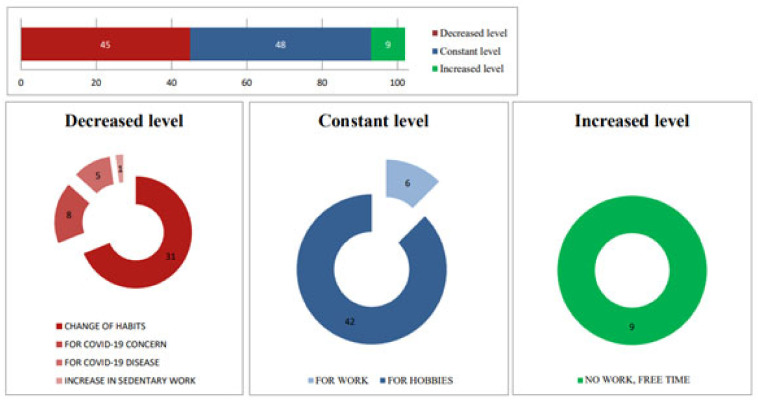
Qualitative questionnaire responses to the physical activity levels during the COVID-19 period.

**Table 1 medicina-60-01624-t001:** Sample characteristics.

Gender	Participants (*n*)	Percentage (%)	Average Age	Average Number of Family Members	Employment Status % (*n*)
Male	73	70.87%	68	2.69	Employed: 50.7% (*n* = 37) Retired: 43.8% (*n* = 32) Unemployed: 5.5% (*n* = 4)
Female	30	29.13%	67	2.69	Employed: 43.3% (*n* = 13) Retired: 50.0% (*n* = 15) Unemployed: 6.7% (*n* = 2)

**Table 2 medicina-60-01624-t002:** Overall summary statistics PACIC questionnaire.

Item	Sample (*n*)	Mean	SD
1	103	1.84	1.45
2	103	2.58	1.69
3	103	4.03	1.40
4	103	4.20	1.19
5	103	3.62	1.62
6	103	2.34	1.61
7	103	4.19	1.42
8	103	4.16	1.53
9	103	1.45	1.12
10	103	2.63	1.73
11	103	3.83	1.50
12	103	3.93	1.52
13	103	2.96	1.69
14	103	2.55	1.72
15	103	2.19	1.70
16	103	1.73	1.42
17	103	2.76	1.80
18	103	1.97	1.56
19	103	1.59	1.26
20	103	1.73	1.34
21	103	2.17	1.62
22	103	2.27	1.66
23	103	4.37	1.31
24	103	2.71	1.70
25	103	1.99	1.50
26	103	2.31	1.34

Legend. SD: standard deviation.

**Table 3 medicina-60-01624-t003:** Overall summary statistics of the Medi-Lite Questionnaire.

Items	Sample (*n*)	Mean	SD
Fruit	103	2.58	1.18
Vegetables	103	0.88	0.45
Legumes	103	1.04	0.64
Cereals	103	1.97	0.17
Fish	103	1.07	0.51
Meat and meat products	103	1.19	0.60
Dairy products	103	1.14	0.71
Alcohol	103	1.34	0.59
Olive oil	103	1.89	0.34

Legend. SD: standard deviation.

## Data Availability

The data supporting this research are available upon request from the corresponding authors for data protection reasons.

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
