# Peer review of "Impact of the COVID-19 Pandemic on Lifestyle Behavior and Clinical Care Pathway Management in Type 2 Diabetes: A Retrospective Cross-Sectional Study"

_medicina, 2024, doi:10.3390/medicina60101624_

Round 1

Reviewer 1 Report

Comments and Suggestions for Authors

I was glad to review the authors' work regarding this fascinating manuscript on the Impact of the COVID-19 Pandemic on Lifestyle Behavior and Clinical Care Pathway Management in Type 2 Diabetes. The manuscript is well-written and the topic is very interesting.

I can consider acceptance for publication of the paper after minor changes.

1) "The coronavirus disease outbreak caused by severe acute respiratory syndrome coronavirus 2 (SARS-CoV-2) has become a global health problem. COVID-19 is a highly infectious and multifaceted disease, and already millions of people have been infected worldwide. According to the literature, the COVID-19 pandemic caused a major reduction of emergency surgical operations as well as overall admissions to emergency departments because of the widespread hospital fear and anxiety experienced by most patients during the peak of this outbreak"

I would suggest adding this important information and consider citing the related articles:

https://pubmed.ncbi.nlm.nih.gov/33986894/

2) I would suggest a brief discussion on the impact of COVID-19 on diabetic patients visiting the emergency department 

3) What about their outcomes when operated on in comparison with non-diabetic patients

4) Did you collect data such as BMI, and ASA scores from the patients of the current study? It would be interesting to have this information as well and see the difference on Lifestyle Behavior and Clinical Care Pathway Management based on BMI and ASA scores.

5) It is also important to categorize the patients of the current study by a degree of treatment control: Good (HbA1c < 7%), Intermediate (7% ≤ HbA1c < 8%), and Poor (8% ≤ HbA1c).

Author Response

Dear Reviewer,

the reply in the file annex.

Thank you again for your time and kindly suggestions.

The Authors

Reviewer 2 Report

Comments and Suggestions for Authors

Dear Editor and Authors,

The authors conducted a retrospective, cross-sectional study using questionnaires in 103 T2DM patients to assess the perceptions of patients with T2D regarding the quality of care they received during the COVID-19 pandemic, adherence to the Mediterranean diet, and the impact of COVID-19 on dietary and physical activity behaviors. Overall, the paper is very well structured. The presentation of methods and results is quite clear and understandable. However, there are still a few points that need to be corrected. First of all, the English language should be revised throughout the paper. Spelling and grammatical errors should be corrected. In my opinion, the weakest points of the study are the small sample size, the fact that the study was conducted in a single center, the number of women participating in the study was much lower than men, and most importantly, we do not know how these investigated parameters affect patients' blood glucose regulation, weight gain, and laboratory results. In addition, there is no control group that allows us to compare how the COVID-19 pandemic affects care, physical activity behaviors, and diet in groups other than T2DM with T2DM patients. As can be expected, unfortunately, this epidemic had an impact on a large number of chronic diseases to a greater or lesser extent. However, we do not have sufficient evidence to determine which patients are the most vulnerable among these. The authors should expand on the limitations of the study and provide guidance on future study designs. As the authors also mention, recall bias is an important problem due to the nature of retrospective studies. The introduction could be written a little shorter. Finally, the same expressions were written twice between lines 303-316. Please edit this. 

I wish the authors success.

Comments on the Quality of English Language

English language should be revised throughout the paper. Spelling and grammatical errors should be corrected.

Author Response

Dear Reviewer,

the reply in the file annex.

Thank you again for your time and valuable commnts.

The Authors

Reviewer 3 Report

Comments and Suggestions for Authors

The topic is interesting and the paper is quite well written. The article covers a current topic. I have some comments:

1) Abstract. Results: During the research period, out of 130 subjects who met all enrollment criteria, 103 patients 32 were included in the study (79.23%). The results obtained from the administered questionnaires 33 were heterogeneous. Social and economic restrictions played an important role in shaping environ- 34 mental habits among this frail cohort of T2D patients, especially in the most stressful period of the 35 pandemic. Please underline the most important results to support the conclusions. 

2) Abstract. Conclusions: healthcare systems must prioritize comprehensive care plans for T2D that 36 address not only physical health but also emotional and social well-being. Post-pandemic, promot- 37 ing healthier lifestyles and empowering patients to manage their condition is crucial. Abstract might be beneficial to include a sentence that briefly summarizes the key findings of the study. This can provide readers with a quick overview of the research. 

3) 1. Introduction 45 The COVID-19 pandemic, declared by the World Health Organization (WHO) on 46 March 11, 2020, and concluded on May 5, 2023, resulted in more than seven hundred mil- 47 lion infections and possibly seven million deaths (as of April 13, 2024). Over about four 48 years since its global spread, it reached dimensions that posed significant challenges for 49 public and social health management in recent history [1,2]. I suggest to improve this part. I invite authors to discuss and add some references, such as:

Quantitative Computed Tomography Lung COVID Scores with Laboratory Markers: Utilization to Predict Rapid Progression and Monitor Longitudinal Changes in Patients with Coronavirus 2019 (COVID-19) Pneumonia. Biomedicines. 2024;12(1):120. Published 2024 Jan 6. doi:10.3390/biomedicines12010120

Radiological-pathological signatures of patients with COVID-19-related pneumomediastinum: is there a role for the Sonic hedgehog and Wnt5a pathways?. ERJ Open Res. 2021;7(3):00346-2021. Published 2021 Aug 23. doi:10.1183/23120541.00346-2021

Beyond the Acute Phase: Long-Term Impact of COVID-19 on Functional Capacity and Prothrombotic Risk—A Pilot Study. Medicina 202460, 51. https://doi.org/10.3390/medicina60010051

4) The lockdown due to the COVID-19 pandemic, despite its challenges in managing 89 therapeutic interventions, offered, in some cases, an unexpected opportunity to improve 90 lifestyle habits. A study conducted in France [49] revealed that a significant number of 91 people with diabetes and overweight used this period to adopt a healthier lifestyle, lose 92 weight, and improve their diet. I suggest to underline the aim of the study and the novelty of this paper.

5) 2.7. Statistical Analysis 166 The statistical analyses for the PACIC and Medi-Lite questionnaires were conducted 167 using Python, focusing on both descriptive and inferential statistics. The analysis of 168 PACIC scores, which assess patients' perceptions of the care they receive for chronic ill- 169 ness management, was subdivided according to the 5A subscales: Assess, Advise, Agree, 170 Assist, and Arrange. Descriptive statistics, such as the mean and standard deviation (SD), 171 were calculated for the PACIC items. These statistics provided a comprehensive overview 172 of the central tendency and variability of responses within the entire sample and across 173 stratified groups. Similarly, the Medi-Lite scores, which reflect adherence to the Mediter- 174 ranean diet, were analyzed by stratifying the dataset according to demographic variables. 175 For each stratification variable (e.g., gender, education level), the counts of Medi-Lite ad- 176 herence levels (categorized as 0 [Low], 1 [Moderate], and 2 [High]) were computed for 177 each dietary item (e.g., vegetables, legumes, cereals). For both questionnaires, inferential 178 statistical tests were performed to explore the relationships between demographic factors 179 and the overall scores. These tests included t-tests or non-parametric alternatives, depend- 180 ing on the data distribution, to assess the significance of differences between groups. Please, underline the most important statistical tests to support the results.

6) 3. Results 183 184 3.1. Characteristics of the Sample 185 Initially, 130 patients attending routine visits were considered. Of these, 20 did not 186 meet the inclusion criteria. Among the 110 eligible patients, 5 did not give consent for the 187 processing of personal data for scientific purposes. Out of the 105 participants who started 188 the study, 2 did not fully complete the questionnaires. Therefore, the final sample in- 189 cluded in our study consisted of 103 patients (all Caucasian), which represents 79.23% of 190 the initial sample (Figure 1). Please, underline in the text the most important statistically sugnificant data to support the results and the conclusions.

7) Discussion. This study aimed to explore the influence of the COVID-19 pandemic on the 407 management of T2D, focusing on dietary and lifestyle factors. Our findings highlight the 408 significant challenges and difficulties in managing T2D in an extremely vulnerable cohort 409 during the pandemic. Even with first-choice pharmacological interventions, preventing 410 more serious health consequences proved challenging for patients with T2D during this 411 delicate period [56]. Please, the discussion section needs to be improved.  It could be interesting to improve the description of the results obtained and compare them with published literature.

8) 5. Conclusions 482 The results of this study emphasize the critical need for a comprehensive approach 483 to managing chronic diseases—one that extends beyond direct medical treatment to 484 include initiatives aimed at fostering healthy lifestyles and active patient participation. 485 Furthermore, these findings underscore the importance of creating tailored interventions 486 designed to mitigate the detrimental effects of crises, such as lockdowns .. Please, underline the novelty of the study and the possible clinical implications.

Comments on the Quality of English Language

Minor changes of English language are required.

Author Response

Dear Reviewer,

the reply in the file annex.

Thank you again for your time and valid suggestions.

The authors

Round 2

Reviewer 3 Report

Comments and Suggestions for Authors

The topic covered is interesting. The authors adequately answered my questions. No further comments